# Variable Seepage Meter Efficiency in High-Permeability Settings

**Donald O. Rosenberry** [1],*, **José Manuel Nieto López** [2], **Richard M. T. Webb** [1] and **Sascha Müller** [3]

1   U.S. Geological Survey, Water Mission Area, Denver, CO 80225, USA; rmwebb@usgs.gov
2   Department of Geology, Faculty of Sciences, University of Malaga, 29071 Malaga, Spain; nietolopezjm@uma.es
3   Department of Geosciences and Natural Resource Management, University of Copenhagen, 1350 Copenhagen, Denmark; samu@ign.ku.dk
*   Correspondence: rosenber@usgs.gov; Tel.: +1-303-236-4990

**Abstract:** The efficiency of seepage meters, long considered a fixed property associated with the meter design, is not constant in highly permeable sediments. Instead, efficiency varies substantially with seepage bag fullness, duration of bag attachment, depth of meter insertion into the sediments, and seepage velocity. Tests conducted in a seepage test tank filled with isotropic sand with a hydraulic conductivity of about 60 m/d indicate that seepage meter efficiency varies widely and decreases unpredictably when the volume of the seepage bag is greater than about 65 to 70 percent full or less than about 15 to 20 percent full. Seepage generally decreases with duration of bag attachment even when operated in the mid-range of bag fullness. Stopping flow through the seepage meter during bag attachment or removal also results in a decrease in meter efficiency. Numerical modeling indicates efficiency is inversely related to hydraulic conductivity in highly permeable sediments. An efficiency close to 1 for a meter installed in sediment with a hydraulic conductivity of 1 m/d decreases to about 60 and then 10 percent when hydraulic conductivity is increased to 10 and 100 m/d, respectively. These large efficiency reductions apply only to high-permeability settings, such as wave- or tidally washed coarse sand or gravel, or fluvial settings with an actively mobile sand or gravel bed, where low resistance to flow through the porous media allows bypass flow around the seepage cylinder to readily occur. In more typical settings, much greater resistance to bypass flow suppresses small changes in meter resistance during inflation or deflation of seepage bags.

**Keywords:** seepage meter; groundwater–surface–water interaction; sediment–water interface

## 1. Introduction

Seepage meters are the only device that can directly quantify exchange between groundwater and surface water. All other methods require measurement of an indirect but related parameter, such as hydraulic gradient, hydraulic conductivity, chemical constituents, or temperature [1]. Although they directly quantify flow of water across a portion of the sediment–water interface isolated by the seepage cylinder, they do so with some resistance to flow that is often not quantified or acknowledged. Therefore, all seepage meter measurements that are not adjusted with an efficiency compensating multiplier under-measure to some extent the actual rates of exchange between groundwater and surface water.

Seepage meters measure the volume of water that flows across a portion of the sediment–water interface bounded by an open-ended seepage cylinder installed in the bed of a surface-water body. The bottom edge of the cylinder extends into the sediment some distance beneath the sediment–water

interface and the closed top extends above the interface at least a few cm but needs to be fully submerged in the water body in which the meter is installed. Commonly, a seepage bag is used to quantify the flow of water across the interface covered by the seepage cylinder. The bag, containing a known volume of water, is attached to the cylinder and the change in volume during the time the bag is attached provides a volume per time (*Q*), which can be normalized to a seepage flux (*q*) by dividing by the area of the sediment–water interface covered by the seepage cylinder:

$$q = E_S \cdot (\Delta V / \Delta t)/A \tag{1}$$

where

$E_S$ is the multiplier to compensate for seepage meter undermeasurement (unitless) and is the inverse of seepage meter efficiency;

$\Delta V$ is the change in volume of water contained in the seepage bag (m$^3$);

$\Delta t$ is the duration of seepage bag connection to the seepage cylinder (commonly minute or day);

*A* is the inside area of the seepage cylinder, equal to the area of the sediment bed being measured (m$^2$).

If water flows upward across the sediment bed, the bag gains in volume; the bag loses water if downward flow occurs. Several design modifications have reduced or eliminated various sources of error, improving both the accuracy and repeatability of measurements and extending the use of the device to a broader range of settings [1].

All seepage meters create some resistance to flow in making a seepage measurement, resulting in a measured rate of flow that is less than what would flow through an undisturbed setting. Reduced seepage during the measurement is caused by numerous factors, including redirecting flow through the various meter components, friction loss at the meter surfaces and from routing flow through small-diameter valves and connection hardware, and inflation or deflation of a seepage bag. Reduced flow through the meter implies that some water is flowing around the meter to discharge beyond the confines of the seepage cylinder. Here, we call this bypass flow. The ratio of the measured to actual seepage is termed the meter efficiency, which can only be determined by installing a seepage meter inside a controlled environment, commonly called a seepage tank or calibration tank. These devices usually consist of a large container partially filled with sediment and with a column of water above the sand sufficiently deep such that a seepage meter can be submerged when it is installed on the sediment surface (e.g., [2–7]). A known rate of flow is created through the sediment to which flow through the seepage meter is compared.

## 1.1. Sources of Inefficiency

Efficiency varies substantially based on the meter design. Early seepage meters routed flow through glass or plastic tubing that extended through a rubber stopper inserted into the seepage cylinder [8–12], the small diameter of which generated substantial friction loss that was directly proportional to the volumetric seepage rate and inversely proportional to the tubing diameter according to the Hagen-Poiseuille relation

$$\Delta P = (8\mu L Q)/(\pi R^4) \tag{2}$$

where

$\Delta P$ is pressure drop over the length of the tubing (kg/m/s$^2$);

$\mu$ is dynamic viscosity determined at the water temperature (kg/m/s);

*L* is the tubing length (m);

*Q* is the volumetric seepage rate (m$^3$/s);

*R* is the radius of the tubing (m).

This can be equated to hydraulic head by dividing by water density ($\rho$) and gravity (*g*), such that

$$\Delta h = (8\mu L Q)/(\pi R^4 \rho g) = (8.16 \times 10^{-4} \mu L Q)/(\pi R^4) \tag{3}$$

where $\Delta h$ is change in hydraulic head of a column of water (m).

Early studies indicated flow resistance (hydraulic head) of about 1 to nearly 60 mm attributed to flow through the connector tube with inside diameters ranging from 5 to 1 mm, respectively [11]. Subsequent work with larger-diameter tubing indicated flow resistance ranging from 4 to 19 mm when a very fast seepage rate of 259 cm/d was routed through tubing ranging in diameter from 7.9 to 4.8 mm, respectively [13]. Later designs routed water from the seepage cylinder to a bag placed inside a seepage bag shelter that was located some distance from the cylinder, primarily to eliminate velocity head effects caused by waves and currents [14–18], but also to move the seepage bag away from the seepage cylinder [1].

Flow resistance, expressed here in terms of hydraulic head and referred to as head, was found to be related primarily to the thickness of the seepage bag material. One study indicated about 2 to 25 mm of head was required to fill a bag with a material thickness of 150 μm, but generally smaller head of about 2 to 5 mm was required to fill a bag with material thickness of 25 μm [15]. Head varied substantially for the thicker plastic bag due to kinks in the bag and bag fullness. Others also indicated that head loss was proportional to bag fullness, with head ranging between 0.1 and 1 mm when the bags were operated between about 1/10 and 3/4 full [19]. Several studies [11,20–22] reported use of much thicker bags that created even greater resistance. Several of the thicker bags were used because they had a built-in neck that made bag attachment to tubes and hardware easier. A laboratory study prior to seepage work on a river in California [6] found bag resistance to be much greater for solar shower and urine collection bags than for the 25 μm thick bag used by Murdoch and Kelly [15]. Other studies used wine bladders and backpack hydration bags [23] or oven-basting bags [16] because of the bag connection convenience, but none provided an associated bag efficiency.

### 1.2. Reported Seepage Meter Efficiency

Seepage meter designs and use have improved substantially since the half-barrel seepage meter was introduced [9], including a general improvement in reported measurement efficiency (Figure 1). Use of larger-diameter hardware for routing water between the seepage cylinder and seepage collection bag and use of less resistant seepage bags [2] has resulted in reported efficiencies as large as 0.95. Reported efficiency values commonly are assumed to be constant and are generally assumed to be controlled by the design of the seepage device.

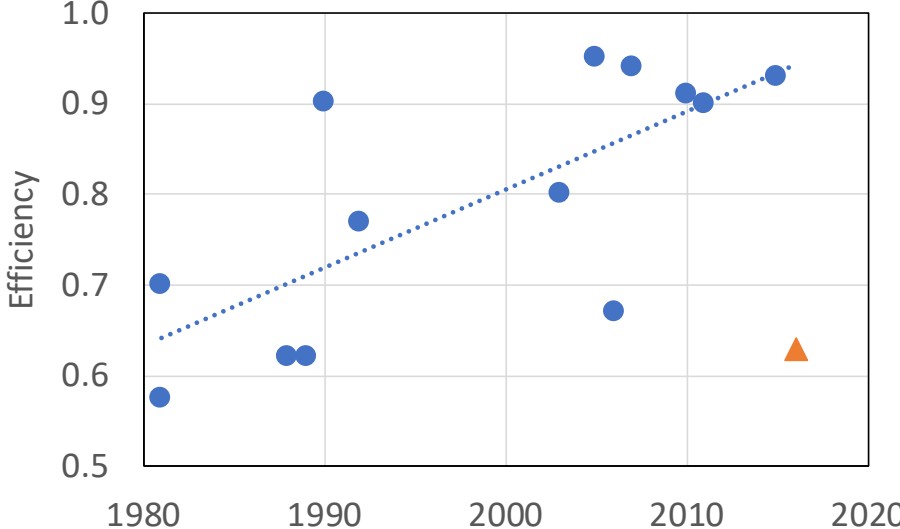

**Figure 1.** Increase in the literature-based seepage meter efficiency determined by relating measured versus known seepage rates in a calibration tank [3,5–7,12,15,19,24–28]. Orange triangle symbol is from Solder et al. [29].

However, as indicated earlier, seepage meter efficiency can vary depending on several factors other than the meter design. An example of the effect of hydraulic conductivity on efficiency can be seen from the data in Figure 1. Identical seepage meters yielded the largest efficiency value of 0.95 and also a much smaller value of 0.63 (orange triangle) when tested in the same seepage meter calibration tank. In the first determination of efficiency [26], the tank was filled with playground sand with a horizontal hydraulic conductivity ($K_h$) of 15 m/d and a vertical hydraulic conductivity ($K_v$) of about 7 m/d [30]. Substantial heterogeneity of flow through the playground sand was subsequently observed [2] and others also have noted that it is very difficult to create a seepage tank that does not have substantial heterogeneity of flow distributed across the sediment–water interface in the tank [3,9,31]. To reduce heterogeneity, the playground sand was replaced with sand from the St. Peter formation [32], silica sand of such uniform sphericity and grain size it commonly is used as a proppant for hydraulic fracturing [33]. Sand grains were well-rounded with a grain diameter varying over a narrow range of 0.45 to 0.55 mm, leading to the assumption that heterogeneity would be greatly reduced with such a uniform porous medium. $K_h$ of the St. Peter sand was measured at 5 different locations in the sand tank and ranged from 48 to 71 m/d and averaged 58 m/d [34,35]. Constant head permeameter measurements (e.g., [36,37]) were made at the same locations, with a shallow insertion depth of 5 cm to provide values of $K_v$ at the sand surface. Those values ranged from 45 to 76 m/d and averaged 58 m/d, indicating that the St. Peter sandstone was virtually isotropic.

Following sand replacement, a separate determination of efficiency was made for an identical half-barrel seepage meter installed in the high-$K$ St. Peter sand as part of a study to test a new tube seepage meter [29]. When deployed in the much more permeable St. Peter sand, seepage meter efficiency decreased from 0.95 to 0.63. Much smaller resistance to flow through the coarser and more uniform St. Peter sand allowed a substantially greater bypass flow around the seepage meter.

These two efficiency determinations of the same design of seepage meter in two different sediment types indicate that efficiency of a seepage meter is not a fixed value but varies substantially depending on the hydraulic conductivity of the sediments in which the device is installed. Given that large-diameter tubing and fittings were used, the seepage bag was likely the largest factor in reduced efficiency with increased sediment permeability.

### 1.3. Purpose and Scope

Use of an automated seepage meter installed in a controlled-seepage tank allows detailed testing of seepage bag and, therefore, manual seepage meter efficiency. Previous studies have related seepage meter bag resistance to either a conductance term or head loss. Here, we have the capability to measure seepage to or from a seepage bag 12 times a minute by attaching the bag directly to an electromagnetic seepage meter (ESM, described in 2.1) installed in the seepage tank. We can then compare measured seepage rates averaged during each minute of bag attachment to flow through the ESM when no bag is attached. Rather than providing one time-integrated value of efficiency for each seepage bag attachment, we can determine efficiency values during every minute of each bag attachment. Measurements are made in highly permeable sediments that provide greater sensitivity and variability in efficiency response during both filling and emptying of seepage bags.

To investigate the effect of hydraulic conductivity on meter efficiency, we used a laboratory-determined seepage meter efficiency at a known hydraulic conductivity to determine a meter-dependent flow resistance. We then numerically simulated bypass flow in three dimensions caused by a simulated meter installation over a range of simulated hydraulic conductivities, keeping the meter resistance constant. For each simulated value of $K$, we simulated bypass associated with three different depths of cylinder insertion into the sediment. We focused our efforts on higher-permeability sediments to be relevant to more recent investigations that have measured faster seepage rates, particularly in high-$K$ hyporheic settings [38–42].

## 2. Materials and Methods

### 2.1. Seepage Bag Efficiency Tests

The effects of variable resistance associated with filling and emptying seepage bags are amplified when conducted in the highly permeable St. Peter sand inside the previously mentioned seepage calibration tank [2]. As mentioned above, earlier studies related bag resistance to changes in bag conductance, defined as the ratio of $Q/h$ [15], or changes in the measured head [19] required to create flow as bags were filled. Here, we quantify changing bag resistance during both filling and emptying of seepage bags by attaching a seepage bag directly to the exhaust port (or intake port if flow is downward) of the ESM while measuring $Q$ through the ESM before, during, and following each bag attachment. We calculate bag efficiency during each minute of bag attachment by averaging flow measured by the ESM 12 times a minute and dividing that value by the average of ESM-measured flow both before and after the bag was attached.

The ESM consists of an electromagnetic flowmeter connected to a large, 1.06 m diameter seepage cylinder that was installed in a 1.5 m diameter seepage calibration tank filled with St. Peter sand (Figure 2). The seepage cylinder covered 50 percent of the total area of the sediment bed in the tank. This larger-diameter cylinder improved the low-flow measurement capability of the ESM, allowing repeatable minimum-flow measurements as slow as about 0.5 cm/d.

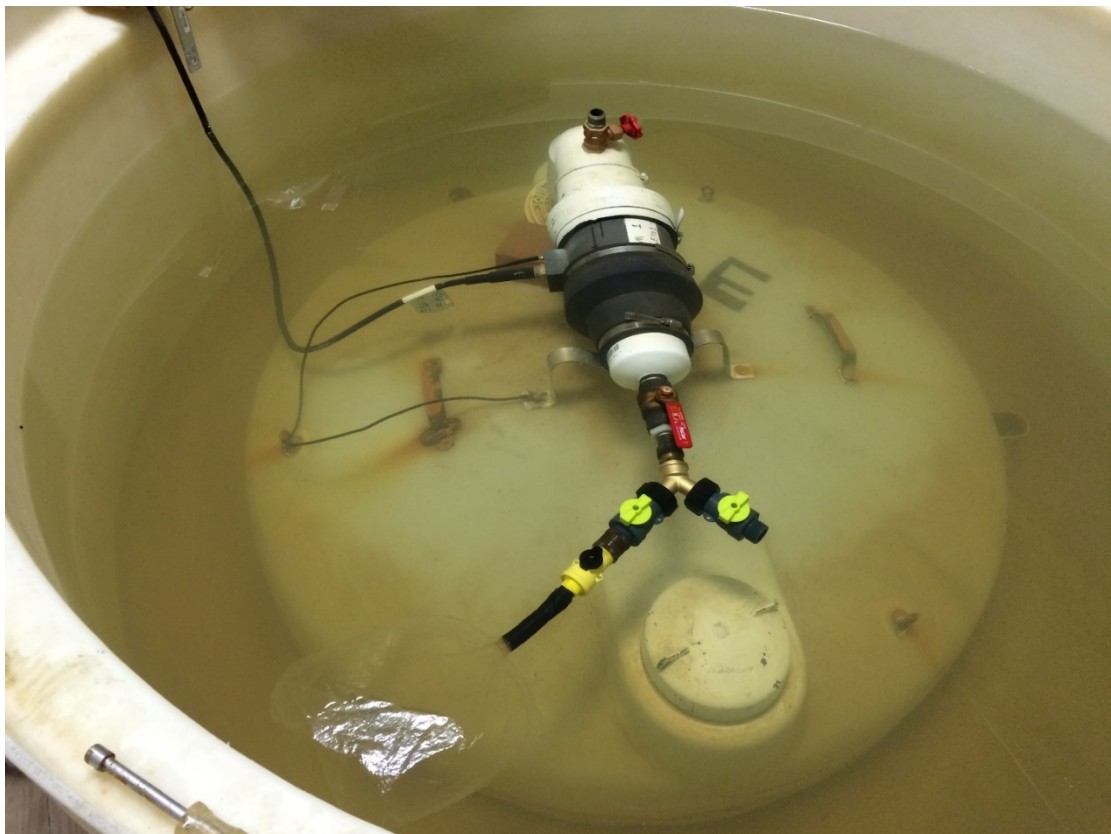

**Figure 2.** Electromagnetic seepage meter (ESM) installed in a 1.5 m diameter seepage calibration tank. A seepage bag is attached at the exhaust end of an electromagnetic flowmeter (gray cylinder with data cable attached). The entire system is submerged. Note that the Y-valve connector with the valve in line with the bag open and the other valve closed.

An electromagnetic flowmeter is based on Faraday's law of induction, which states that a voltage will be induced in a conductive fluid (water) that is directly proportional to the fluid velocity as it passes through an electromagnetic field. The electromagnetic flowmeter has no moving parts and has

no components that extend into the cylindrical flow field, making it an ideal sensor for quantifying slow flows [13]. An electromagnetic flowmeter designed to measure flow in boreholes was modified by Quantum Engineering Corporation, Loudon, TN, USA, for use as a seepage meter [43]. (Any use of trade, firm, or product names is for descriptive purposes only and does not imply endorsement by the U.S. Government.) The flowmeter outputs voltage when a fluid passes through the 1.3 cm diameter open cylinder at the center of the electromagnetic field created by the sensor. Voltage is generated at a rate of approximately 1 mV per 1 mL/min of flow. Sensor output is scanned every 5 s by a digital datalogger programmed to average 12 values each minute. During prolonged constant flow, the standard deviation of the 5 s data averaged each minute was 0.7 mL/min. For fast seepage rates, bags would fill or empty in just a few minutes, resulting in only a few measurements of efficiency for each test. In those situations, data were averaged once each 15 s instead of once a minute to provide four times the data density. Standard deviation for sensor output averaged over 15-s periods was 3.4 mL/min. Flow in ml/min ($Q$) was converted to flux ($q$) in terms of cm/d (the most commonly used units in the seepage meter literature) by dividing by the 8938-cm$^2$ area covered by the seepage cylinder and multiplying by 1440 min in each day. The resulting values for standard deviation of ESM data in units of cm/d averaged over 1 min or 15 s are 0.1 or 0.5, respectively.

A pulse-count flowmeter was used to indicate total flow through the seepage tank. Data from the pulse-count flowmeter indicated a standard deviation of 0.5 mL/min, similar to the 0.7 mL/min standard deviation for data from the ESM.

For each bag efficiency determination, averages of stabilized ESM values immediately before and after the duration of bag attachment to the ESM were used as a reference to which flow during bag attachment was compared. Examples of data collected during 6 separate seepage bag attachments are shown in Figure 3 by the shaded rectangles indicating the duration of each bag attachment to the ESM exit/entry port. Values are negative because downward flow was occurring in the seepage tank. Larger negative values during the last two bag connections indicate greater resistance to flow because the bag was less than 25 percent full. (This same series of measurements is presented in more detail in the section on upward versus downward flow.).

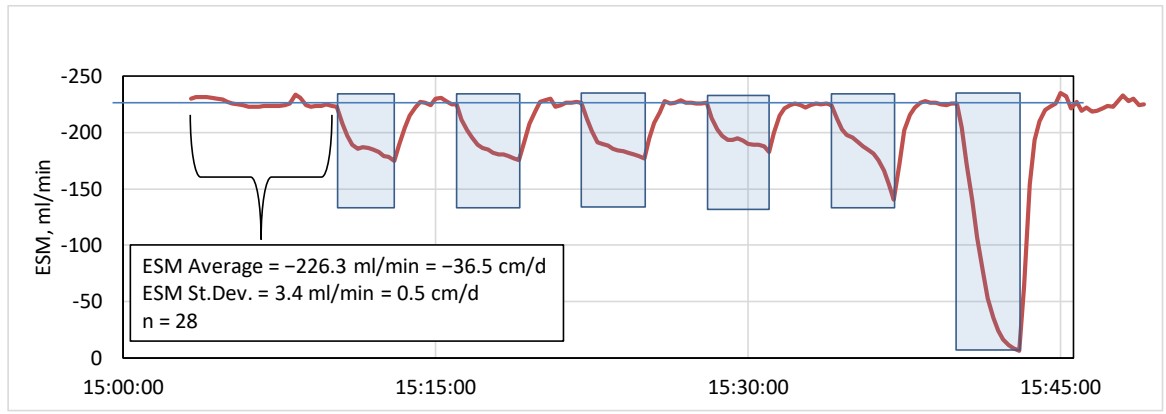

**Figure 3.** ESM output in ml/min with average values provided every 15 s. Blue rectangles are periods during seepage bag attachment efficiency measurements. Horizontal blue line indicates the average ESM value when no bag is attached. ESM average, standard deviation, and n are for periods with no bag attachment when output was stabilized.

### 2.2. Simulation of K Versus Seepage Meter Efficiency

The U.S. Geological Survey (USGS) software MODFLOW 2005 version 1.12.00 [44] was used to simulate bypass flow around a seepage meter extending into a porous medium directly beneath the sediment–water interface. The model domain was 5 m by 5 m along the x and y axes. Cell size within 2 m of the vertical model boundaries was 0.1 m by 0.1 m. Within the central 1 m by 1 m area of the model, cell size was reduced to 0.025 m for greater resolution of flow around a centrally located,

simulated 0.57 m diameter seepage cylinder. Along the z (vertical) axis, the 1.01 m extent of the model domain was divided into 13 layers. Eight model layers were each 0.1 m thick from 0 to 0.8 m. Above those, four model layers were each 0.05 m thick from 0.8 to 1.0 m, both to provide better resolution of flow around the simulated seepage cylinder and also to allow simulation of variable cylinder insertion depths into the flow domain. The top 0.01 m thick layer was created to allow $K$ of the 432 cells in the top layer inside the simulated seepage cylinder to be reduced to represent total resistance of flow through a seepage meter. The circular sidewall of a seepage cylinder inserted into a porous medium was simulated using the horizontal flow barrier (HFB) feature that is part of MODFLOW. The HFB feature was set to have a thickness of 1 mm and a horizontal $K$ of $1 \times 10^{-5}$ m/d along 1 or 2 of the sides of the square cell walls that collectively formed the approximate circle representing the sidewall of the seepage cylinder. The HFB extended 0.06, 0.11, or 0.16 m from the top of the flow field to simulate cylinder-installation depths of 0.05, 0.1, or 0.15 m into the sediment bed.

The vertical sides of the model domain were assigned as no-flow boundaries and the top and bottom of the model were assigned as constant head boundaries. Head of the bottom boundary was set at 0.1 m larger than the top boundary. Sediment was simulated as isotropic with an initial $K$ of 1 m/d, typical of sandy settings and similar to $K$ when the seepage calibration tank was filled with playground sand. With these initial conditions, vertical flow was simulated at a rate of 0.1 m/d, typical of moderately fast seepage in many natural settings [45] and similar to many of the seepage rates during seepage meter calibration tests conducted in the seepage calibration tank [2].

Hydraulic conductivity along the vertical axis ($K_z$) of the top layer within the simulated seepage cylinder ($K_{seep}$) was iteratively reduced until the average of vertical flow through the area of the simulated seepage cylinder ($q_{seep}$) was close to either 90 or 95 percent of the average of vertical flow through the rest of the domain ($q_{bulk}$), simulating seepage meters with a noted efficiency of 90 or 95 percent. $K$ for all other cells within the flow domain ($K_{bulk}$) was set at 1 m/d during the initial determination of fixed values for $K_{seep}$. Once those values of 0.160 m/d for 90 percent efficiency and 0.252 m/d for 95 percent efficiency were determined, $K_{seep}$ was kept the same while $K_{bulk}$ was varied by orders of magnitude from 0.01 to 100 m/d to determine both the variations in meter efficiency relative to $K$ and also the volumetric extent of bypass flow around the seepage cylinder. Although $K_{seep}$ was fixed, $K_x$ and $K_y$ of the top layer inside of the simulated seepage cylinder varied and were always set equal to $K_{bulk}$. Three separate runs at each value of $K_{bulk}$ were made to simulate insertion depths of 0.05, 0.10, and 0.15 m.

## 3. Results

Reductions of seepage meter efficiency related to the seepage bag and to depth of seepage-cylinder insertion are presented from experiments conducted in the seepage meter calibration tank. Sensitivity of measured seepage related to sediment hydraulic conductivity is then demonstrated through results of the MODFLOW simulations. All data collected in the seepage tank are available at [46] and data generated by MODFLOW simulations are available at [47].

*3.1. Efficiency Results from Seepage Calibration Tank Measurements*

3.1.1. Bag Efficiency Related to Bag Fullness

Bag efficiency related to bag fullness was tested over a range of seepage rates from 2.1 cm/d (Figure 4A) to 38.7 cm/d (Figure 4F). Efficiency decreased continuously during most bag attachments independent of initial bag fullness, but the best efficiencies occurred for bags that were operated in approximately the mid-range of the 3-L bag capacity (Figure 4, solid lines). Variance of bag efficiency was much larger during slow seepage (Figure 4A,B), even accounting for the longer bag-attachment times. Bag efficiency decreased rapidly and non-linearly during the early portions of most bag connections, particularly when the bag was beyond about 75 percent full upon attachment (Figure 4A,C). An offsetting effect sometimes occurred when bags that were nearly empty were attached

during slow upward seepage. Within the first 5 to 10 min, flow to the bag was increased substantially compared to the pre-connection rate (Figure 4A–C). This initial large seepage flux also was reported by early adopters of seepage meters (e.g., [48]) and is caused by the seepage bag not being in a relaxed position. During all but one of the highly anomalous initial seepage rates, the bag was nearly empty (or nearly full, as shown later) at the beginning of the measurement. During some tests, efficiency reductions followed by increased bag efficiency likely was due to folds or creases in the bag being overwhelmed as water was added to the bag. This feature is shown in Figure 4D, 65–72 percent fullness, where decreasing bag efficiency during the first 9 min was followed by increasing bag efficiency until the end of the test. Collectively, results show a "best-fullness" operating range of about 15 to 60 percent. This fullness range is slightly lower than the 25 to 75 percent fullness recommended by Murdoch and Kelly [15].

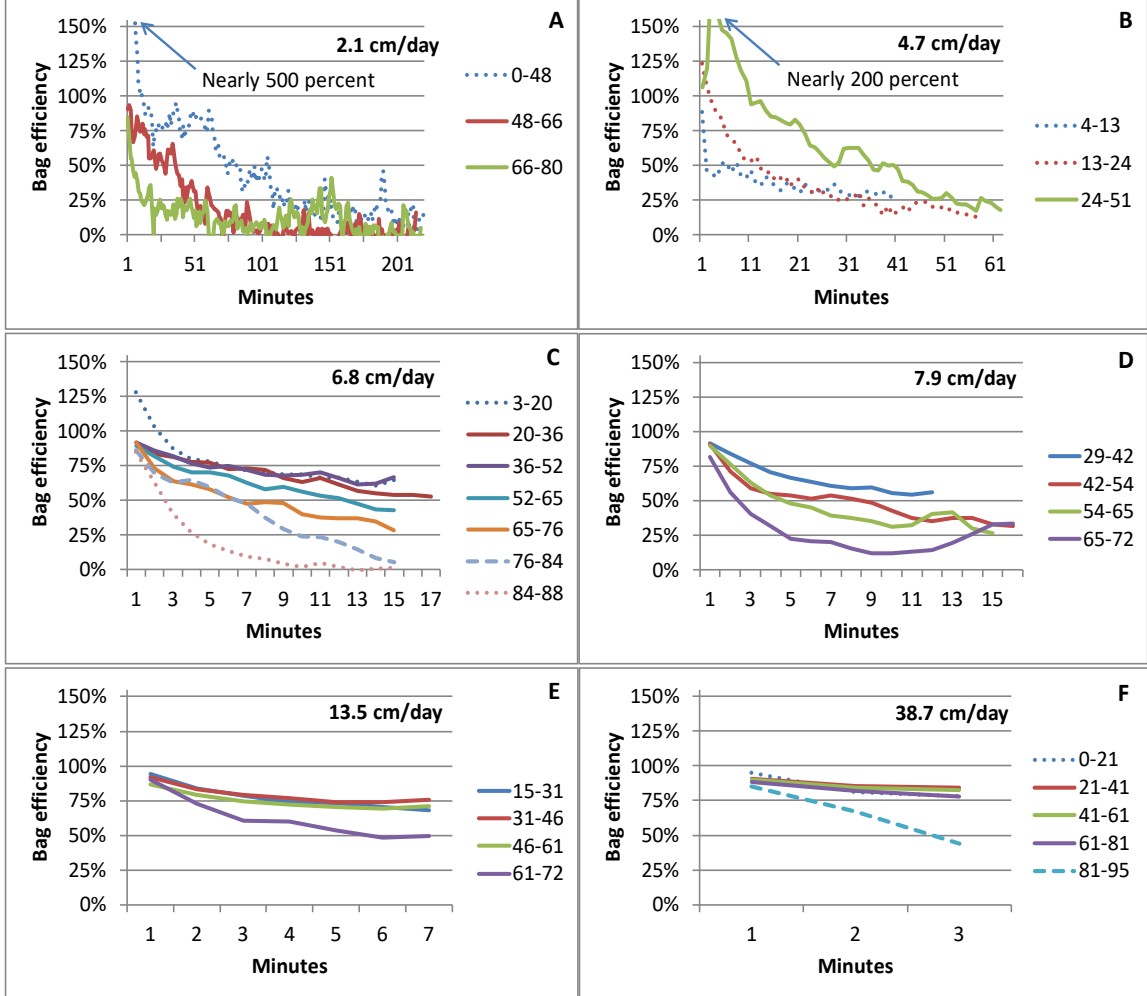

**Figure 4.** Effect of bag fullness on bag efficiency during upward seepage. Bag efficiency is plotted for each minute the bag was attached to the ESM. Bag-attachment plots are labeled based on the percent fullness of the bag at the beginning and ending of each attachment (i.e., 0–48 indicates the bag was 0 percent full at the beginning of the attachment and 48 percent full at the end of the attachment). Panels (**A–F**) represent: tests conducted with seepage rates ranging from 2.1 to 38.7 cm/d, respectively. Values in bold in each chart are average seepage rates through the ESM (cm/d) when no bag was attached. Dotted or dashed lines indicate tests when bag fullness was largely outside of the recommended 25 to 75 percent.

### 3.1.2. Efficiency Related to Seepage Velocity

Seepage meter efficiency generally increased with increasing seepage velocity. Considering only the data collected when bag fullness was within the recommended mid-range, average seepage meter efficiency based on data presented in Figure 4 ranged from 18 percent for slow seepage to 87 percent for fast seepage (Table 1). Only the data with bag-attachment fullness ranging from 48 to 66 percent were included from the 2.1 cm/d plot (Figure 4A), and values >100 were not included from the 4.7 cm/d efficiency plot (Figure 4B). Variation in efficiency from one bag attachment to another was also much smaller for faster seepage rates.

**Table 1.** Seepage rate, in cm/d, relative to bag efficiency, in percent; averaged over each bag connection (second column) or averaged over all bag attachments that were considered in the acceptable mid-range of bag fullness (third column). For averages of first 3 min of bag attachment (fourth column), attachments with initial values >100 percent were not included.

| Seepage Rate, cm/d | Bag-Averaged Efficiency, % | Average Efficiency, % | Average Efficiency for First 3 Min, % |
|---|---|---|---|
| 2.1 | 18 | 18 | 81 |
| 4.7 | 51 | 51 | 59 |
| 6.8 | 68, 73, 62, 51 | 64 | 79 |
| 7.9 | 67, 49, 46 | 53 | 74 |
| 13.5 | 78, 79, 75 | 77 | 80 |
| 38.7 | 88, 86, 87 | 87 | 87 |

### 3.1.3. Efficiency Related to Bag Connection Duration

Efficiency was much improved if only the first 3 min of bag attachments shown in Figure 4 are considered (Table 1, right-most column). Efficiency values range from 59 to 87 percent and a slight relation between seepage velocity and efficiency is indicated, with a best-fit linear model explaining only 1/3 of the variance.

Longer rather than shorter bag connection times have generally been recommended [1,2] because they allow a better time averaging of any short-duration perturbations, such as inadvertent touching of the bag during measurement or occasional increases or decreases in bag resistance due to creation or reduction of folds or curves in the bag material. Longer bag connection times also average the effects of initial large seepage rates related to attachment of a nearly empty bag, as previously mentioned [48,49]. However, longer-duration bag connections in the seepage test tank resulted in larger decreases in bag efficiency until a very small or virtually zero bag efficiency is reached (Figure 5). Bag attachments all started either very close to empty (Figure 5A,C) for upward seepage or close to full (Figure 5B,D) for downward seepage, and bags were attached until at least 75 percent of the bag capacity was filled or emptied (except for Figure 5B, when about 70 percent of the bag volume was lost). Resulting bag efficiency measurements were highly variable. Variance was particularly large for slow seepage (Figure 5A) and decreased substantially when seepage was faster (Figure 5C). The only data that indicated reasonably stable efficiency was for fast seepage (Figure 5C,D). The chart for fast downward seepage indicates a sharp decrease in efficiency after about 13 to 15 min of bag attachment, a decrease that continued until efficiency essentially reached zero after about 22 min (Figure 5D). This feature may also have been displayed for upward seepage if those bag attachments had extended for longer durations. Use of a larger-volume seepage bag also resulted in poor efficiency.

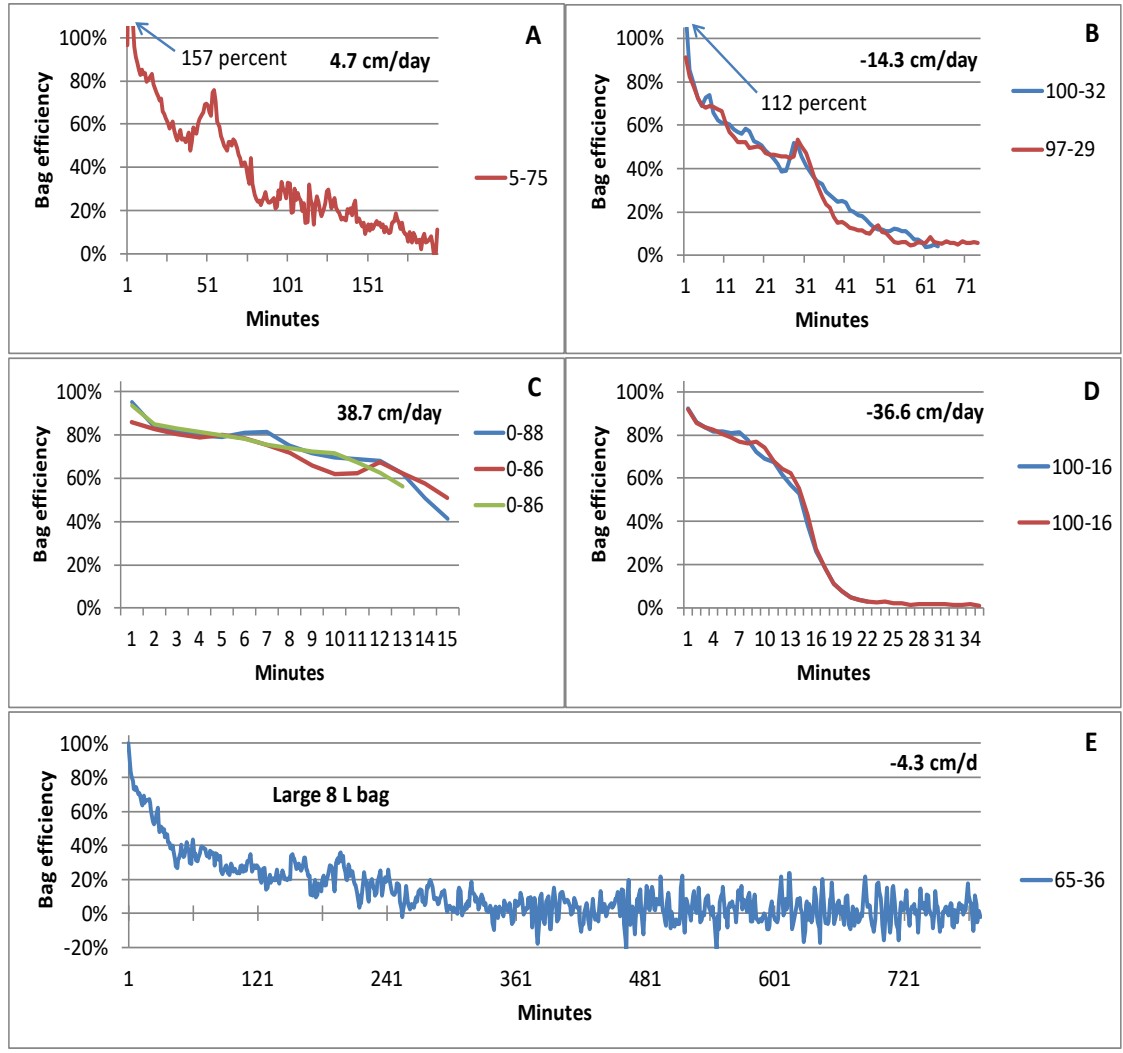

**Figure 5.** Seepage bag efficiency for long-duration bag connection times. Panels (**A**,**C**) show upward seepage (empty-to-full bag connections) and (**B**,**D**) downward seepage (full-to-empty bag connections). Panel (**E**) is a large-volume seepage bag. Legend entries indicate beginning and ending percent bag fullness. Values in bold in each chart are seepage rates through the ESM (cm/d) in between bag attachments.

A larger-capacity 8 L bag 65 percent full was connected for 13 h during slow downward seepage (Figure 5E). Efficiency decreased rapidly to about 40 percent during the first 45 min and then decreased more slowly until reaching virtually zero flow after about 5.5 to 6 h. Very little additional water flowed from the bag during the remaining 6 to 7 h, even though 2970 mL of water remained in the bag upon removal. Others also have indicated reduced data quality with large seepage bags because the inner surfaces of the bag were stuck together due to adhesion, essentially blocking portions of the bag from providing water to the bag outlet [27].

### 3.1.4. Efficiency Related to Upward vs. Downward Seepage

In theory, bag resistance should not depend on direction of flow to or from the bag. However, early results indicated substantially larger efficiency values for increased than for decreased volume in a seepage bag [12], leading to the notion that seepage meters work better for upward than for downward seepage. Flow through the highly permeable sand in the seepage calibration tank does not indicate a consistent bias (Figure 6). For slowest flows, bag efficiency was slightly larger for upward

seepage. The average of two bag attachments during upward flow, with bag fullness ranging from 13 to 51 percent, indicated 51 percent efficiency, whereas the average of three bag attachments during downward flow, with bag fullness ranging from 80 to 22 percent, indicated an average efficiency of 45 percent. For moderate seepage, where ESM seepage values were +7.9 and −14.3 cm/d, the average bag efficiency for upward flow was slightly smaller than for downward flow, at 58 and 63 percent, respectively. The results were also similar for comparisons at the fastest seepage rates. When ESM values during upward and downward seepage were +38.3 and −36.4 cm/d, bag efficiencies for upward and downward seepage were 76 and 84 percent, respectively. The only consistent characteristic for bag efficiency is the aforementioned increase in bag efficiency with increased seepage rate, no matter the flow direction (Figure 6).

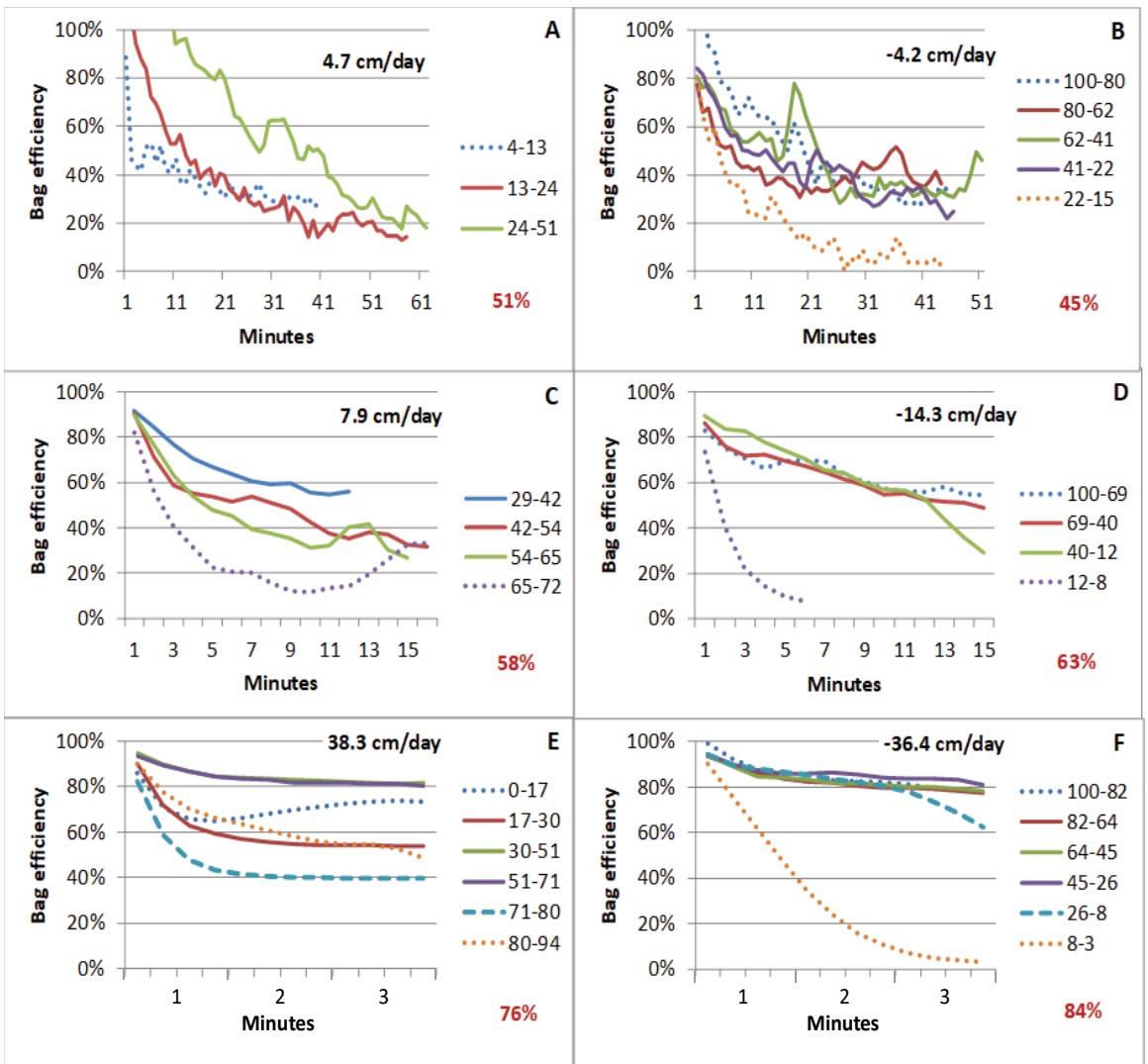

**Figure 6.** Seepage bag efficiency for upward versus downward seepage. Panels (**A,C,E**) show upward seepage at slow, moderate, and fast seepage rates, respectively. Panels (**B,D,F**) show downward seepage at slow, moderate, and fast seepage rates, respectively. Legend entries indicate beginning and ending percent bag fullness. Values in bold in each chart are seepage rates through the ESM (cm/d) in between bag attachments. Percentages indicated in red are average bag efficiencies for bag tests when bag fullness was within the recommended operating range.

### 3.1.5. Inertial Effects of Shutting Down Seepage Prior to a Measurement

All efficiency measurements were made while connecting and removing bags from a Y connector containing two ball valves, one on each leg of the Y (Figure 2). This allowed one valve to remain open to maintain flow through the meter while a seepage bag was connected or removed from the other leg on the Y connector, providing uninterrupted flow through the ESM and quicker stabilization of ESM output through the elimination of sudden, large, artificial changes in flow. This maintenance of continuous flow rarely if ever has been done for most manual seepage meter measurements. Typically, flow through the area covered by the seepage meter is stopped for a minute or more while the bag is connected and properly positioned within the bag shelter, the shelter lid moved into place, and then the valve on the seepage bag is opened at a convenient or prescribed time. In the seepage tank, one leg of the Y valve was always open, either when a bag measurement was being made or when a bag was in the process of being attached or removed.

Two sets of bag efficiency measurements were made, one with and one without the use of the Y connectors to compare the influence of maintaining constant flow versus the normal procedure of interrupting flow during bag attachment and removal. Data indicate that substantial reductions in seepage can occur if flow is stopped to attach or remove a seepage bag. Seepage measured in the traditional way averaged only 59 percent of the measured rate when the Y valves allowed continuous flow through the meter (Table 2). If only data with bag fullness in the recommended mid-range are used, the ratio of stopped/continuous flow is 0.53. Furthermore, the average of output from the ESM in between periods of bag attachment was −2.7 cm/d when the Y valves were not used and −4.2 cm/d when they were used, a ratio of 64 percent, indicating that flow did not quickly or fully return to normal during the 5 to 7 min in between each bag-attachment period. This observation is considered more fully in the Discussion section.

**Table 2.** Seepage bag results, in ml/min, during uninterrupted (Continuous flow) and during standard (Stopped flow) bag attachments. Tank flow was set to −5 cm/d (31 mL/min through the ESM) for all measurements.

| Percent Bag Fullness | Continuous Flow, mL/min | Stopped Flow, mL/min | Stopped ÷ Continuous |
|:---:|:---:|:---:|:---:|
| 75–100 | −15.02 | −11.93 | 79% |
| 25–75 | −12.00 | −7.31 | 61% |
| 25–75 | −12.67 | −7.54 | 60% |
| 25–75 | −12.00 | −4.09 | 34% |
| 0–25 | −5.09 | −3.11 | 61% |
| Average | −11.36 | −6.80 | 59% |

### 3.1.6. Seepage Cylinder Installation Depth

Modeling of bypass flow, presented in the next section, indicates that seepage meter efficiency increases with increased cylinder insertion depth. This condition also was tested in the seepage tank by comparing bag measurements made before and then after pressing the seepage cylinder deeper into the sand, increasing the insertion depth from 4 to 8 cm. The bag measurement made at the 4-cm insertion depth began with a large initial efficiency of 123 percent, declined rapidly and nonlinearly, and averaged 37 percent (Figure 7). After pressing the meter to 8 cm depth, measurement efficiency still decreased with time. However, the decrease was more linear and efficiency averaged 54 percent, indicating an increase in efficiency with increased installation depth.

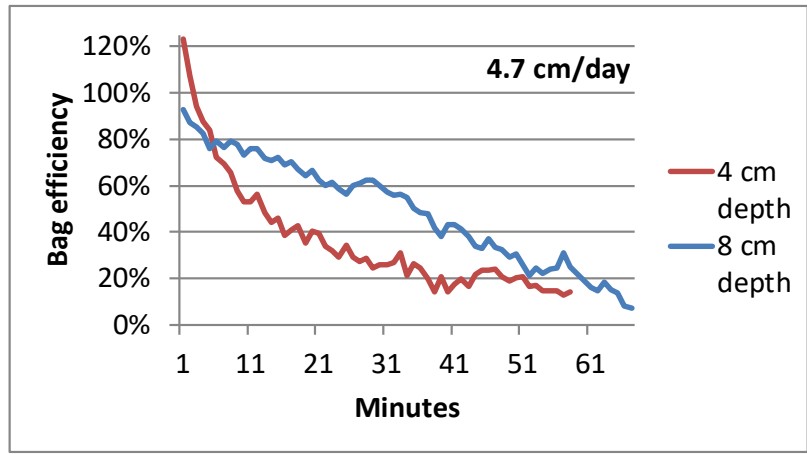

**Figure 7.** Two bag efficiency measurements with a seepage rate of 4.7 cm/day, the first made with the seepage cylinder inserted 4 cm into the sand bed and the second made with the seepage cylinder inserted 8 cm into the sand.

### 3.2. Influence of Hydraulic Conductivity on Meter Efficiency

Modeling results indicate that seepage meter efficiency decreases substantially once $K$ gets above about 1 m/d (Figure 8). Even when simulating a highly efficient seepage meter determined in the laboratory to have an efficiency of 95 percent when $K$ is 1 m/d, that efficiency decreases to about 60 percent when $K$ is 10 m/d, and decreases to about 10 percent when $K$ is 100 m/d. Efficiency also increases slightly with increasing seepage-cylinder insertion depth with the biggest range in efficiency, varying from 52 to 62 percent, occurring at $K$ = 10 m/d. (Figure 8).

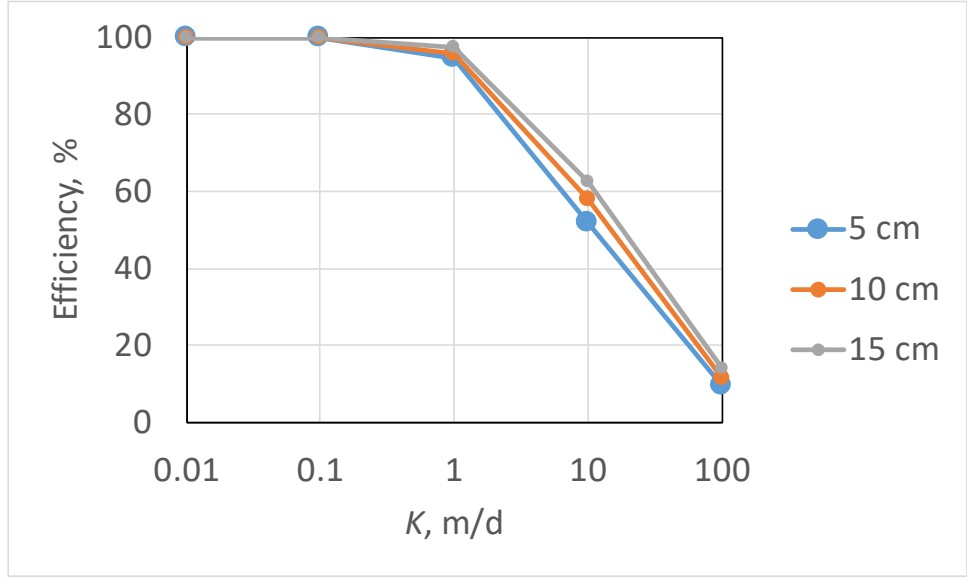

**Figure 8.** Simulated seepage meter efficiency for meter insertions at 5, 10, and 15 cm depths over a range of $K$ from 0.01 to 100 m/d.

Bypass flow that increases with decreased efficiency can be visualized using MODPATH, a particle-tracking package within the MODFLOW software domain. Traces of 80 "particles", released at the bottom of the flow domain along the central 3 m length of a row of cells that passed through the center of the simulated seepage cylinder, collectively show the deflection of flowlines that represent bypass flow around a seepage cylinder installed at a depth of 0.1 m (Figure 9). No deflection of flowlines was indicated for $K_{bulk}$ of 0.1 m/d (or for $K_{bulk}$ = 0.01 m/d—results not shown in the figure).

Bypass flow is limited to within about 0.01 m of the cylinder walls when $K_{bulk}$ = 1 m/d. Bypass flow extends to about 0.04 m from the cylinder walls when $K_{bulk}$ is increased to 10 m/d and extends to nearly 0.08 m from the cylinder walls when $K_{bulk}$ is increased to 100 m/d (Figure 9).

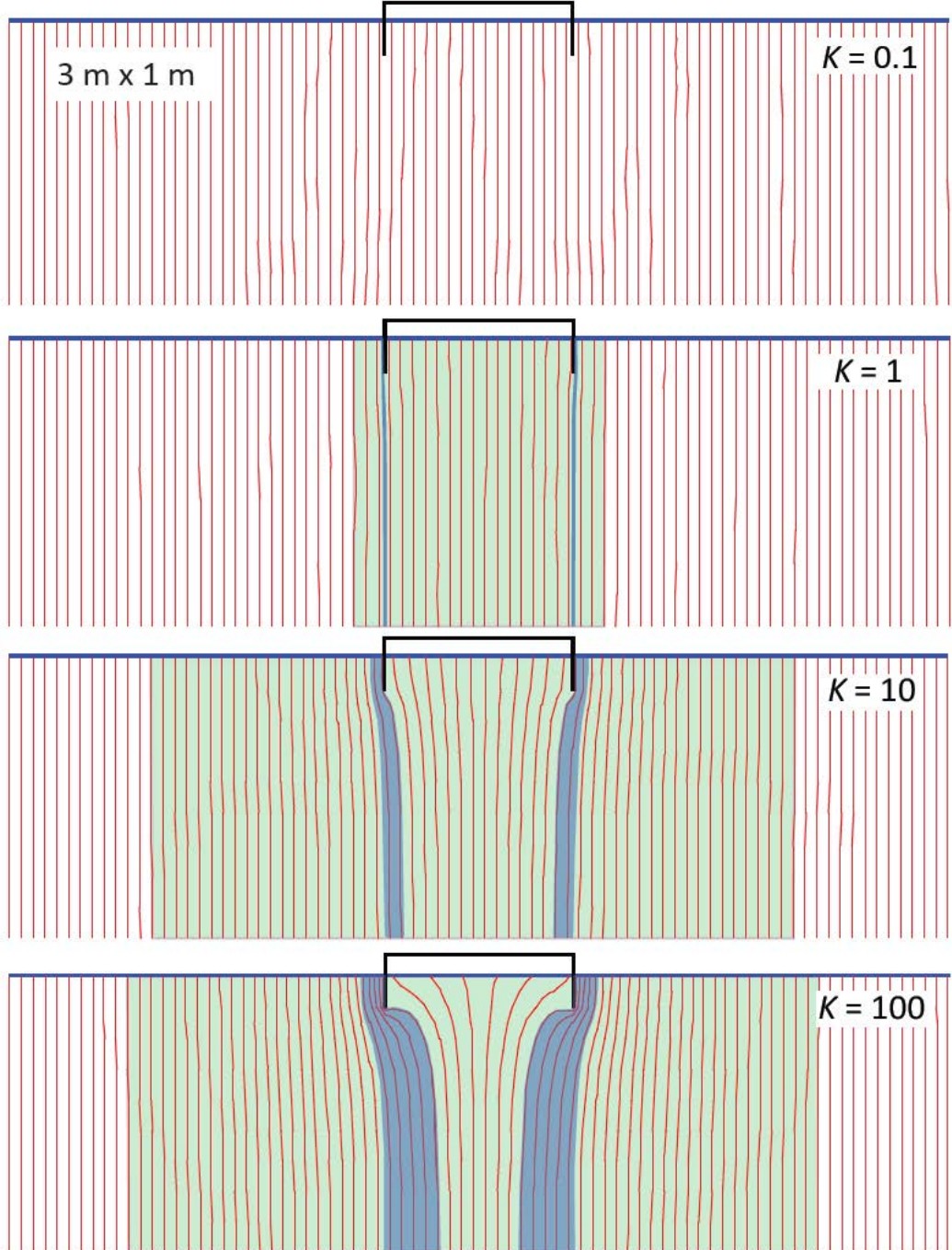

**Figure 9.** Deflection of flow around a simulated seepage meter inserted 0.1 m into the flow domain with a 95 percent meter efficiency at $K_{bulk}$ = 1 m/d. Simulations shown for $K_{bulk}$ of 0.1, 1, 10, and 100 m/d. Red lines indicate flow paths of 80 "particles" released at the bottom of the domain. Blue shading indicates bypass flow of water that would otherwise have discharged within the simulated confines of the seepage cylinder. Green shading indicates area where otherwise vertical flow is visibly diverted laterally by the bypass flow.

The velocity of bypass seepage leaving the flow domain is increased as it is diverted around the confines of the seepage cylinder. MODFLOW results indicated that the fastest seepage rates were adjacent to the exterior of the simulated cylinder walls and exceeded the median seepage rate at the top of the model domain by 2, 14, and 29 percent for $K_{bulk}$ of 1, 10, and 100 m/d, respectively.

The flow domain beyond the area of bypass flow is also altered somewhat, with compressed flowlines and slightly increased local seepage velocity. MODPATH flowlines were visibly deflected from vertical (Figure 9, green-shaded area) to distances of 0.1, 0.7, and 0.8 m for $K_{bulk}$ of 1, 10, and 100 m/d, respectively.

## 4. Discussion

Seepage meter efficiency clearly depends on the hydraulic conductivity of the sediment in high-*K* settings. The large reduction in efficiency based on 3-d flow modeling is consistent with similar modeling that related efficiency to pressure head. The reduction in efficiency determined here, from 95 to 60 to 10 percent for *K* values of 1, 10, and 100 m/d, is similar to pressure head-determined efficiency of 90, 40, and 5 percent for approximately the same range in *K* [31]. Furthermore, the model-determined reduction in efficiency, from 95 to 40 percent when *K* is increased from 1 to 10 m/d, is similar to the measured efficiency reduction (95 to 63 percent, (Figure 1)) when $K_v$ of the sediment in the seepage tank increased from 7 to about 70 m/d. Although the message is disturbing and indicates widely varying efficiency values, both based on the sediment type and based on duration of bag attachment, the bag efficiency data from the seepage tank presented here represent the worst case. Only a few types of natural settings have such highly permeable sediments, such as high-energy beaches or mobile coarse-sand or gravel beds in rivers. Other settings typically have sediments with orders-of-magnitude lower *K*. For all those settings, the literature-based seepage meter efficiency values are likely to be adequate. Once the seepage meter resistance is less than that of the sediment in which it is installed, the seepage meters function with their stated efficiency. Nevertheless, lessons learned from this work can lead to better practices that will generate higher-quality data in all settings.

### 4.1. Recommended Practices to Maximize Efficiency and Consistency

Prior to this study, all determinations of seepage meter efficiency were based on the full duration of each bag attachment. Data presented here provide efficiency values for each minute of each bag attachment period, with 15 s resolution for short bag attachment durations. Based on these high-temporal-resolution data when seepage bags were connected to the ESM, the following recommendations can be made for high-*K* settings:

- Operate seepage bags in the mid-range of their capacity;
- Use shorter rather than longer bag connection times;
- Maintain continuous flow through the meter prior, during, and after bag attachment;
- Use thin-wall, pliable seepage bags and avoid using high-capacity seepage bags;
- Press the seepage cylinder more deeply into the sediment where possible;
- Avoid routing water through sharp bends in the seepage-measurement device;
- Use a meter that does not require inflation or deflation of a seepage bag.

Each of these recommendations is discussed further in the sections below.

### 4.1.1. Operate Seepage Bags from about 1/5 to 2/3 Full

Earlier studies had indicated that bags operated between about $\frac{1}{4}$ and $\frac{3}{4}$ full would result in better and more consistent data collection [15,48]. Here, reasonably high efficiency values were obtained with bag fullness as small as about 15 to 20 percent, but large efficiency reductions occurred during several runs with bag fullness as small as 65 to 70 percent.

Avoiding attaching bags that are nearly full or empty eliminates most of the spikes in seepage upon first opening of the bag valve (e.g., Figure 4A,C; Figure 5A). During those spikes, the bag evidently

was not at equilibrium pressure, which created much faster seepage than would be warranted by the rate of flow in the seepage tank. The initial spike in flow through the bag occasionally was well more than double the actual seepage rate, but the effect was usually gone within the first 5 to 10 min of bag attachment. However, for some efficiency measurements, those initial large flows during the first few minutes provided a substantial offset to a subsequent decrease in bag efficiency as the period of bag attachment continued. A good example is shown in Figure 4B, where the bag attachment with the initial efficiency values larger than 100 percent had subsequent efficiency values that continued to be large during the rest of the bag attachment, resulting in a bag-averaged efficiency of 65 percent (51 percent if values > 100 are discounted) versus 37 percent for the previous bag attachment (red dotted line). Those offsetting processes could result in a better time-averaged bag efficiency if not for the initial large flows being so inconsistent.

4.1.2. Use Shorter rather than Longer Bag Connection Times in Highly Permeable Sediments

What was quite consistent, though, was the steady decrease in bag efficiency during nearly all bag attachments, no matter whether bags were decreasing or increasing in volume. Only the tests conducted with the fastest seepage rates indicated a relatively consistent efficiency. Therefore, if the seepage bag is attached for a shorter rather than longer time, the time-averaged seepage rate will have a higher associated measurement efficiency. This is counter to earlier recommendations for longer bag-attachment times that would better average initial aberrant seepage rates [1]. Shorter bag connection times should not be so short that the change in volume in the bag is within or close to the measurement error.

As mentioned earlier, this disturbingly large sensitivity to bag-attachment duration only applies to high-$K$ settings. Furthermore, the initial, artificially fast seepage rates occur primarily when the seepage bag is operated out of its mid-range of bag fullness, which can easily be avoided. In more typical sediments where $K$ is less than about 1 m/d, longer bag connection times are still preferred and will better time-average and minimize any brief inadvertent perturbations to a seepage bag [1].

There may be multiple causes for the steady decrease in bag efficiency reported here and that others also have documented [15,19]. First, as a seepage bag fills or empties, the relaxed state of the bag, no matter the initial fullness, easily accommodates a change in volume until the bag membrane needs to move substantially and more uniformly to accommodate additional filling. As the smallest wrinkles and folds are removed during early filling of a bag, any continued filling eventually requires wholesale expansion of the bag surface in all directions, creating a greater resistance to continued filling and a greater reduction in bag efficiency. Evidently the same process also occurs as the bag deflates and additional resistance is presented in the creation of folds and wrinkles during the shrinkage of the bag volume.

A second, largely inertial, process that is somewhat independent of the seepage bag also may be important in particularly coarse-grained settings. For several of the series of seepage bag tests, there was a small, but detectable and relatively consistent reduction in flow through the ESM during the times in between sequential periods of bag connection. Following each bag attachment, when flow would presumably return to the pre-bag-attachment rate, the new equilibrium flow rate was slightly smaller. These reductions in "between-attachment" flows were small, and largely linear. For 6 different series of bag connection tests, the between-attachment flow decreased at rates ranging from 2 to 7 percent per hour and averaged 4 percent per hour. Therefore, flow just prior to the beginning of the next bag attachment would be slightly smaller than during the previous bag attachment. The implication is that some of the additional bypass flow created by the added resistance of the attachment of a seepage bag is maintained for at least the several minutes in between bag attachments.

This process may also occur simply due to the presence of the seepage meter on the sediment bed, even if a seepage bag is never attached. The seepage tank and ESM with no bag attached were run overnight in between two days of bag efficiency tests. Shortly after initiation of an overnight seepage rate of 9 cm/d in the seepage tank, output from the ESM was indicating flow at a rate of 4.5 cm/d

(Figure 10). While flow in the seepage tank was maintained at a steady rate, flow through the ESM decreased during the night until around 04:30, when the ESM seepage rate stabilized at slightly less than 3.4 cm/d. That stabilized rate continued until 5:45, at which point tank flow was turned off for a zero-flow calibration period. It appears that the small resistance presented by a seepage meter with no bag attached was sufficient to slowly divert an increasing amount of bypass flow around the seepage cylinder. Reduction in seepage occurred at a rate of 3 percent/hour until the new equilibrium flow was reached, more than 8 h later. These results indicate increasing bypass flow on a temporal scale well beyond that of the seepage bag tests conducted in the test tank and indicate a disturbance to flow simply from placing a seepage cylinder on the bed, a feature that has not previously been reported. It also is highly likely that this influence would be so small as to be unmeasurable for installations in sediments with $K$ more typical of most natural settings.

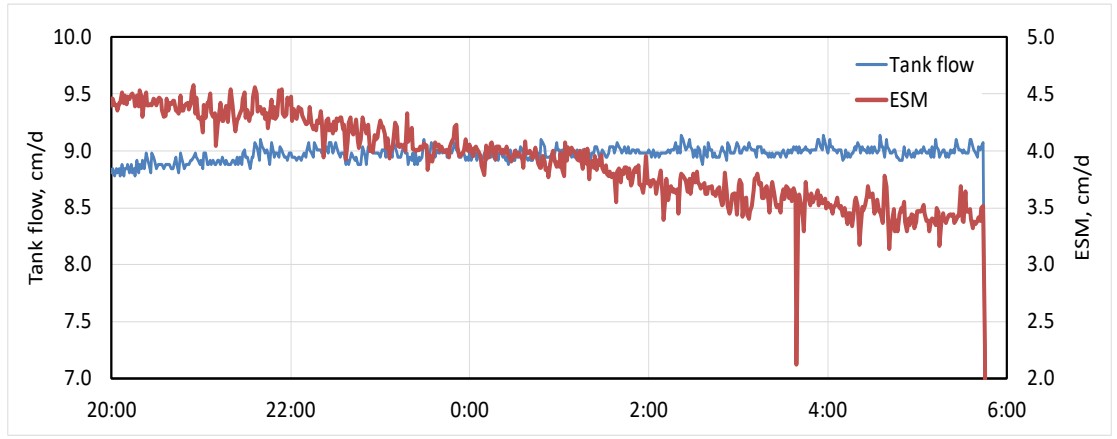

**Figure 10.** Overnight flow through the seepage tank and the ESM in between two sets of seepage bag tests. A zero-flow period was initiated at 5:45. Large reduction in ESM output at 3:39 was likely due to external electromagnetic noise in the building.

### 4.1.3. Maintain Continuous Flow through the Seepage Cylinder during Bag Attachment and Removal

The use of Y valves to prevent interruption of flow through the seepage cylinder was done primarily to eliminate any time-averaging errors associated with the ESM. Although output was recorded every 5 s, noisy data were occasionally observed whenever a sudden change in flow was imposed in the seepage tank, or when a valve connected to the ESM was opened or closed. Use of the Y valves was expected to eliminate potentially noisy data that would be unrelated to the seepage bag. However, because the highly permeable sand greatly increased system sensitivity to any source of resistance, stopping flow prior to and after each bag measurement resulted in a collective and substantial increase in bypass flow and reduction in overall meter efficiency. Although this inertial effect likely is very small in more typical sediments, the process still could occur if bypass flow creates preferential flow paths that do not entirely shut down between bag measurements. The addition of a Y valve is inexpensive, adds little system complexity, and is a small price to pay for improved data integrity.

### 4.1.4. Use Thin-Wall Seepage Bags and Avoid Using a High-Capacity Seepage Bag

The use of thin, flexible seepage bags has previously been suggested [1,6,15] and only the best available seepage bags were used for this study. The large reductions in bag efficiency during nearly all of the bag efficiency tests, as well as the substantial variance in efficiency values for slower seepage rates, indicates that folds or twists in the bag, or simple wholesale expansion of the bag as it nears fullness, can create a large reduction in measured seepage during any given measurement even if the meter design generally is efficient. These deleterious effects are greatly minimized in more typical, lower-$K$ settings.

Only one measurement was made using a large, 8 L thin wall, flexible seepage bag and those results were not promising. At a relatively slow ESM-measured seepage rate of −4.3 cm/d, bag efficiency decreased rapidly to less than 40 percent in the first 40 min and then continued to decrease more slowly over the next 300 min until efficiency was close to zero about 350 min after bag attachment (Figure 5E). Even though flow from the bag was very close to zero from about 350 to 792 min after bag attachment, the bag upon removal still contained nearly 3 L of water and was 36 percent full. Something had blocked flow from the bag during the last 7.4 h of bag attachment. This sample size of only one is supported by others who also have indicated that large bags are more likely to provide poor-quality data. Perhaps, as the bag shrinks, bag surfaces can come into contact due to adhesive forces, or the bag could drift or fold over itself [27]. A larger bag allows less frequent attention to a seepage meter measurement and can be a convenient feature that allows measurements overnight without the bag becoming too full or empty before bag removal. However, there is an increased likelihood that flow will be artificially slowed due to unpredictable movements or resistance to movements of the bag. Furthermore, if the bag is placed inside a bag shelter, which is highly recommended for all seepage meter measurements [1], the shelter will need to be substantially larger, making it more susceptible to movement in response to currents or waves, which adds an additional source of measurement error.

### 4.1.5. Install the Seepage Cylinder Deeper into the Bed Sediment

The proper depth of seepage meter installation has been discussed several times by previous investigators [10,11,14,50] and depends to some extent on the physical setting. Insertion depth typically varies between 5 and 20 cm. For lakes or wetlands or other settings where flow is expected to be primarily vertical, the installation depth only needs to be large enough to ensure a good seal, meaning that there are no gaps beneath the bottom edge of the seepage cylinder where water could flow freely past the wall of the cylinder without flowing through sediment. Most studies suggest an installation depth of 10 to 15 cm. Results presented here indicate that a deeper installation provides better data. However, if meters are installed in hyporheic settings, where the axis of flow is often closer to horizontal than to vertical, a deeper meter installation can block the horizontal component of flow and create a bias in the measured seepage. In those settings, a shallower installation depth is often preferred to more adequately measure hyporheic exchange.

The sand bed in the seepage tank was nearly flat, allowing an initial installation depth of only 4 cm. That shallow installation depth allowed substantial bypass flow even with very small increase in flow resistance within the meter. Once the cylinder was pressed to an insertion depth of 8 cm, the amount of bypass flow decreased and flow through the ESM to or from the seepage bag became more consistent (Figure 7), although efficiency still decreased with time at about the same rate. MODFLOW simulations confirmed the efficacy of a deeper cylinder insertion. Because a deeper insertion depth created a longer flow path for bypass flow, the total increased resistance associated with bypass flow resulted in less water being deflected around the seepage cylinder and greater meter efficiency.

### 4.1.6. Avoid Sharp Bends that Can Create Inertial Effects

The ESM, because of the large-diameter flowmeter and associated hardware and the lack of a seepage bag, was considered to have an efficiency of essentially 1 prior to this work in the high-$K$ sand in the seepage tank. However, comparing flow through the ESM with overall flow through the tank indicated that the ESM efficiency without any attached bag was only 43 percent in the highly permeable St. Peter sand. Resistance due to routing flow through the open cylinder at the center of the flowmeter was very small. Applying the ESM flowmeter open-cylinder diameter and length of 1.3 and 15.2 cm, respectively, to the Hagen-Poiseuille equation modified to solve for head loss (Equation (3)), head loss associated with the fastest flow recorded during seepage tests of 275 mL/min is a very small 0.1 mm. However, additional efficiency loss could be due to angular momentum as vertical flow is turned 90 degrees to flow horizontally through the flowmeter and then through the Y valve before entering or originating from an attached seepage bag (Figure 2). Earlier studies of

routing flow through long reaches of tubing between the seepage bag and seepage cylinder indicated very small reductions in measured seepage with tubing lengths of at least 10 m. However, if the tubing was curved, then seepage reductions were larger [26].

Small reductions in measurement efficiency likely also occur when water flows around bends and folds from a seepage bag that is out of alignment. Placing a bag inside a bag shelter helps to keep the bag in a proper orientation and minimizes that possibility. Water also must flow through the bag shelter, requiring several holes drilled in the shelter to minimize any additional resistance associated with flow through the bag-shelter walls. Seepage bag shelters were not used in the seepage tank because there was no current and the bags maintained proper shape and positioning during all tests, a good example being evident in Figure 2.

### 4.1.7. Replace the Bag with another Flow-Measurement Device

Perhaps the easiest way to improve seepage meter efficiency is to eliminate the seepage bag. In highly permeable settings, even small changes in bag resistance can have a substantial effect in unexpected ways. For example, numerical modeling indicates a small reduction in efficiency with increasing seepage velocity, but data from the seepage calibration tank indicate the opposite (Table 1). One possible explanation is that faster seepage creates greater momentum of water inside the seepage bag. Larger momentum within the bag may allow bag inflation or deflation to continue unabated, overwhelming small changes in bag resistance associated with creation or removal of folds and bends in the bag surface as the bag expands or shrinks during each measurement. This would minimize the observed trend of reduction in efficiency with duration of bag attachment. This added complexity is yet another reason to replace the bag with a flowmeter, particularly in high-permeability settings.

Several types of flowmeters have been used to quantify seepage without the use of a bag [13,19,31,51–57]. It generally is assumed that these devices have very high measurement efficiency in most settings. Most of those devices are either more complex or much more expensive or both. A simple and inexpensive dye displacement design quantifies seepage by measuring movement of a dye through a 9.5 mm diameter transparent submerged tube [19]. Seepage through the dye-displacement tube was double the rate measured by a seepage bag. A new device that simply measures change in head until it reaches equilibrium shows great promise and should have an efficiency of 1 [57].

## 5. Conclusions

Both physical measurements and numerical modeling indicate seepage meter efficiency becomes highly variable and decreases substantially with increased hydraulic conductivity for measurements made in highly permeable settings. Therefore, seepage in settings with hydraulic conductivity greater than about 1 to 5 m/d likely is faster than measured unless efficiency values are adjusted for those types of settings.

Seepage meter efficiency determined during each minute of bag attachment varied substantially and nearly always decreased with increasing measurement duration. Efficiencies near 1 at the beginning of a measurement often were reduced to 0.2 to 0.6 when the seepage bag was removed. The best consistency is achieved when measurements are conducted with the seepage bag ranging from 1/5 to 2/3 full. Better and more consistent meter efficiency also occurs with (1) deeper insertion of the seepage cylinder into the bed sediments, (2) not stopping flow as is commonly done, but instead maintaining constant flow through the seepage cylinder during attachment and removal of seepage bags, and (3) making measurements with shorter rather than longer bag connection durations. This latter recommendation is in contrast with previously suggested longer-duration bag connections but applies only for highly permeable settings. Longer bag connection times are still recommended for lower-permeability sediment more typical of most natural physical settings.

These conclusions, based on both physical measurements and numerical modeling, apply only to highly permeable settings, such as high-energy, wave-washed steep shorelines or fluvial settings with highly mobile sand or gravel beds. For all other physical settings in which seepage meters are more

commonly installed, where effects identified here are greatly minimized, meter efficiency is likely close to what has been reported in the literature. Nevertheless, following the practices recommended here should minimize those likely small influences and generate more consistent and representative data.

**Author Contributions:** Conceptualization, D.O.R.; Methodology, D.O.R., J.M.N.L., S.M.; Software/Modeling, R.M.T.W.; Validation, D.O.R., J.M.N.L., R.M.T.W., S.M.; formal analysis, D.O.R., J.M.N.L., R.M.T.W., S.M.; Investigation, D.O.R., J.M.N.L., R.M.T.W., S.M.; Resources, D.O.R.; Data curation, D.O.R., R.M.T.W.; Writing—original draft preparation, D.O.R.; Writing—review and editing, D.O.R., J.M.N.L., R.M.T.W., S.M.; Visualization, D.O.R., R.M.T.W.; Supervision, D.O.R.; Project administration, D.O.R.; Funding acquisition, D.O.R., J.M.N.L., S.M. All authors have read and agreed to the published version of the manuscript.

**Funding:** This research received no external funding.

**Acknowledgments:** We thank the International Doctorate Exchange Program, Universidad de Málaga, for providing funding that led to this collaborative effort. We also thank Peter Engesgaard and the HOBE Center for Hydrology Hydrological Observatory, University Copenhagen, for intellectual and logistical support. We thank Jim Lundy, Minnesota Department of Health (retired), for his critical comments on an earlier version of the manuscript. We also thank Jim for raising this issue more than a decade ago when he noticed that his seepage rates were faster with shorter bag attachment times. Dismissed at the time as being due to coincidence or measurement technique, it turns out he was well ahead of the rest of us.

**Conflicts of Interest:** The authors declare no conflict of interest.

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
