# Peer review of "Variable Seepage Meter Efficiency in High-Permeability Settings"

_water, doi:10.3390/w12113267_

Round 1

Reviewer 1 Report

General Comments:

This study evaluated the seepage-meter efficiency with considering various factors such as seepage-bag fullness, seepage velocity, duration of bag attachment, direction of seepage flux, and depth of insertion into the sediments. I believe that the massive experiments were well performed and the results were also well discussed. I have several minor comments.

Comments and Suggestions:

  • It would be better to explain the principle of the electromagnetic seepage-meter (ESM).
  • Was the ESM made by the authors? If purchased, the product company name and product specifications should be introduced in the subsection of 2.1.
  • Line 136: The location of reference numbers [3,5-7,12,15,19,25-29] should be moved to the end of the ‘~ in a calibration tank’.
  • In Figs. 4 to 6, the efficiency values were above 100 % near the starting time. The reason should be described in the text. In my experience, it took a very long time to stabilize the seepage-meter settings.
  • The variations of the seepage-meter efficiency affected by the bag fullness, seepage velocity, duration of bag attachment, direction of seepage flux, and installation depth have been well discussed in section 3.1. However, the reduction in efficiency over time was quite steep to be used in stream. Have such features also been found in any other research?
  • It is advisable to add some quantitative results to the conclusions. The current conclusion is too qualitative.

Reviewer 2 Report

see attachment
